



# The instrument constant of sky radiometers (POM-02),
# Part II: Solid view angle
Akihiro Uchiyama[1], Tsuneo Matsunaga[1] , Akihiro Yamazaki[2]
[1] Center for Global Environmental Research, National Institute for Environmental
Studies, Tsukuba, Ibaraki, 305-8506, Japan
[2] Meteorological Research Institute, Japan Meteorological Agency, Tsukuba, Ibaraki,
305-0052, Japan
*Corresponding to*: Uchiyama Akihiro (uchiyama.akihiro@nies.go.jp)
**Abstract**
Ground-based networks have been developed to determine the spatiotemporal
distribution of aerosols using sky radiometers. In this study, errors related to the solid
view angle (SVA) of sky radiometers, which are used by SKYNET, were investigated.
The SVA is calculated using solar disk scan data, the measured radiances around the
solar direction in 0.1 × 0.1 degree increments. These measurements include the
scattered light from aerosol and air molecules, as well as the direct solar irradiance,
causing errors in the SVA calculation. The influence of these errors was evaluated with
simulations. From the results of these simulations, if the aerosol optical thickness is
less than 0.5 at 550 nm and the aerosol does not include large particles, such as desert
dust particles, then its influence on the SVA calculation was less than 0.5%. Problems
with the software for the SVA calculation were also investigated. First, the data
processing does not consider the change of airmass (solar zenith angle) during the
solar disk scan measurement. In practice, if a measurement is made in the period
when the change in airmass is small, then the error is small. Second, before starting
data processing, the minimum measured value is subtracted from the measured values,
resulting in underestimation of the SVA by 1 to 4%. Thirdly, the values between 1.4
and 2.5 degrees are not properly extrapolated, resulting in overestimation of the SVA



by 0.6 to 2.1%. The second and third error sources partially cancel each other out, and
the total error is an underestimation of 0.5 to 1.9% of the actual value. Furthermore,
the annual trend in the SVA was examined. In both the visible and near-infrared
regions, this trend cannot be seen in 4 and 8 years of data, respectively. The seasonal
variation of the SVA was also examined, but no clear seasonal variation could be
detected.

**1. Introduction**
Atmospheric aerosols are an important constituent of the atmosphere. Aerosols
affect not only the global climate through the radiation budget both directly and
indirectly (e.g., Ramanathan et al. 2001, Lohmann and Feichter 2005) but also human
health as one of the main components of air pollution.
Atmospheric aerosols have a large variability in time and space. To measure the
spatio-temporal distribution of aerosols, ground-based observation networks such as
AERONET (AErosol RObotic NETwork) (Holben et al. 1998) and SKYNET (Takamura
and Nakajima 2004) have been developed and extended, and remote sensing methods
from space have been developed using the near-ultraviolet to near-infrared
wavelengths.
For ground-based observations, the solar direct irradiance and sky radiances are
measured, and the aerosol characteristics are retrieved by analyzing these data. To
improve the measurement accuracy, it is important to know the characteristics of the
instrument and to be able to accurately calibrate it.
In SKYNET, radiometers POM-01 and POM-02 manufactured by Prede Co. Ltd.,
Japan are used. These radiometers are called 'sky radiometers', and measure both the
solar direct irradiance and sky radiances. The objectives in this study are to
investigate the current status and issues with sky radiometers.
There are two constants that we must determine to be able to make accurate
measurements. One is the calibration constant. The other is the solid view angle (SVA)
of the radiometer. In Part I (Uchiyama et al. 201#), the temperature dependence of the
sensor output was investigated and the calibration constants determined by the
Improved Langley method and normal Langley method were compared. An alternative
method to determine the calibration constant for the 940 nm channel and the
near-infrared channels (1225, 1627, 2200 nm) was shown using on-site measurement
data.
In Part II, the problem related to the SVA of the sky radiometer is described. The





SVA connects the sensor output to the sky radiance, which has units of
energy/(wavelength)/sr. Overestimation (underestimation) in the SVA leads to
underestimation (overestimation) of the single-scattering albedo (SSA). Therefore, it is
necessary to accurately determine the SVA (Khatri et al. 2016, Hashimoto et al. 2012).
In section 2, the accuracy of the current method for the SVA calculation is
investigated based on simulations. Then, in section 3, we describe the problem with the
current SVA calculation program. This software is attached to the SKYRAD package
(Nakajima et al. 1996), which is used to retrieve aerosol parameters from sky
radiometer data. In section 4, we also show the trend in the SVA and seasonal
variation using the data obtained at MLO and JMA/MRI. In section 5, the results and
conclusions are presented.

**2. Simulation study of SVA estimation error**
The sensor output $V$ when measuring the radiances from the sky with a sky
radiometer can be written as follows:

$$
\begin{aligned}
V &= \int_{\Delta} C(\lambda_0) f(\Omega) I(\Omega) d\Omega \\
  &= C(\lambda_0) \bar{I} \Delta\Omega
\end{aligned}
\tag{1}
$$


where $C$ is the sensitivity, $I(\Omega)$ is the sky radiance in the direction of $\Omega$, $f(\Omega)$ is
the response function of the radiometer field of view,

$$
\bar{I} = \int_{\Delta} f(\Omega) I(\Omega) d\Omega \bigg/ \Delta\Omega
\tag{2}
$$


$$
\Delta\Omega = \int_{\Delta} f(\Omega) d\Omega
\tag{3}
$$


and, for simplicity, the wavelength integration is omitted. Here, $\Delta\Omega$ is the SVA,
which is related to the mean sky radiance in the direction of $\Omega$, and errors in the SVA
result in errors in the retrieved SSA. Therefore, the SVA is an important instrument
parameter.
The SVA can be obtained by integrating the output of parallel light incident on the
radiometer from all directions (see Appendix A). The SVA can also be obtained even if
the light source has a finite size: the SVA can be obtained by integrating the output
obtained while scanning the light source (see Appendix B).
To determine the SVA, a method using the measurement data around the sun was
proposed by Nakajima et al. (1996). The radiances around the direction of the sun in
0.1 × 0.1 degree increments are measured; this is called a "solar disk scan". Using
these data, the SVA is calculated.



An example of measurements of the radiance of the sun and around the sun is shown
in Fig. 1. The measurement at POM-02 (red line) was performed horizontally at
intervals of 0.1 degree scattering angles, where the wavelength is 500 nm. Here,
"horizontally" means that the measurements were performed while keeping the zenith
angle the same as the solar zenith angle. In Fig. 1, the values measured by the image
sensor by shading the solar disk are also shown. Both measurement values are
normalized by the value at a scattering angle of −3 degrees, where a negative value
means the left side is facing the sun. The image sensor can measure up to a scattering
angle of 1 degree. By comparing both measurements, we can see that the output of
POM-02 is affected by the direct solar irradiance for up to about 2.5 degrees from the
sun direction.
The hood of POM-02 is designed so that the full field of view (FOV) is 1 degree. The
size of the sun disk is about 0.5 degrees. Therefore, the direct solar irradiance can
enter the detector for angles up to about 0.75 degrees from the sun's center. However,
the comparison between both measurements shows that the sensor output of POM-02
is affected by the direct solar irradiance for angles up to about 2.5 degrees from the
sun's center.
The cause of the increase in the output is considered to be stray light. Since the
length of the hood and the size of the lens are finite, even if the angle from the sun
center exceeds 0.75 degrees, the direct solar light strikes the lens and results in "stray"
light. This stray light reaches the detector and increases the output, and is smaller
than the measurement of the direct sun by three orders of magnitude or more, but the
integrated value has a magnitude that can affect the estimation of the SVA.
Furthermore, when solar light is used as the light source, aerosols and air molecules
exist between the light source and the instrument. Therefore, the scattered light from
aerosols and air molecules is included in the measurement of the direct solar
irradiance. The influence of this scattered light must also be considered.
As seen from Fig. 1, roughly speaking, the FOV of POM-02 consists of a core part
from 0 to 0.5 degrees and a wing part from 0.5 to 2.5 degrees.

$$\Delta\Omega = \Delta\Omega(core) + \Delta\Omega(wing)$$

$$= \int_{\Delta\Omega(core)} f(\Omega)d\Omega + \int_{\Delta\Omega(wing)} f(\Omega)d\Omega \tag{3}$$

Estimating the magnitudes of the two terms gives the following:



$$\Delta\Omega(core) = \int_{\Delta\Omega(core)} f(\Omega)d\Omega$$

$$\cong \int_{\Delta\Omega(core)} 1 \cdot d\Omega \qquad (4)$$

$$= 2\pi(1 - \cos(0.5\,\mathrm{deg}))$$

$$= 2.39 \times 10^{-4}$$

$$\Delta\Omega(wing) = \int_{\Delta\Omega(wing)} f(\Omega)d\Omega$$

$$\cong \int_{\Delta\Omega(wing)} f_{wing}\,d\Omega \qquad (5)$$

$$= 2\pi(\cos(0.5\,\mathrm{deg}) - \cos(2.5\,\mathrm{deg}))f_{wing}$$

$$= 5.74 \times 10^{-3} f_{wing}$$

As seen from Fig. 1, $f_{wing} \approx 10^{-3}$. Therefore, the ratio of the terms is
$$\frac{\Delta\Omega(wing)}{\Delta\Omega(core)} \approx \frac{5.74 \times 10^{-3} f_{wing}}{2.39 \times 10^{-4}} = 2.4 \times 10^{-2} \qquad (6)$$

This means that neglecting the wing part results in underestimation of the magnitude
of the SVA by about 2%. If $f_{wing} \approx 10^{-2}$, then the contribution of the wing part to the
SVA is about 20%, and the instrument should be repaired. If $f_{wing} \approx 10^{-4}$, then the
contribution is about 0.2%, and the wing part can be ignored.
When the direction of the sun is measured, the sensor output $V(\Omega = 0)$ is as follows:
$$V(\Omega = 0) = C\left[ \int_\Delta f(\Omega')I_0 g(\Omega')d\Omega' + \int_{\Delta\Omega} I_{sca}(\Omega')f(\Omega')d\Omega' \right] \qquad (7)$$

$$= v(0) + C\Delta\Omega\bar{I}_{sca}(0)$$

where
$$v(0) = C\int_\Delta f(\Omega')I_0 g(\Omega')d\Omega' \qquad (8)$$

$$\bar{I}_{sca}(0) = \frac{1}{\Delta\Omega}\int_{\Delta\Omega} I_{sca}(\Omega')f(\Omega')d\Omega' \qquad (9)$$

and $I_0 g(\Omega')$ is the solar radiance distribution. The first term on the right-hand side
of eq. (7) is the contribution of the direct solar irradiance, and the second term is that
of the scattered radiance.



When the direction of the sun is $\Omega = \Omega_0$, the sensor output $V(\Omega = \Omega_0)$ is as
follows:

$$V(\Omega = \Omega_0) = C\left[\int_\Delta f(\Omega_0 + \Omega')I_0 g(\Omega')d\Omega' + \int_{\Delta\Omega} I_{sca}(\Omega_0 + \Omega')f(\Omega')d\Omega'\right] \tag{10}$$

$$= v(\Omega_0) + C\Delta\Omega\bar{I}_{sca}(\Omega_0)$$

where the first term on the right-hand side is the contribution of the direct solar
irradiance, and the second term is the scattered radiance. If $\Omega_0$ is outside of the field
of view, then the first term is zero and only the second term is needed.
Currently, based on the data of the solar disk scan measurement, the SVA is
calculated by the following equation:

$$\Delta\Omega' = \int_{\Delta\Omega} \frac{v(\Omega) + \Delta\Omega C\bar{I}_{sca}(\Omega)}{v(0) + \Delta\Omega C\bar{I}_{sca}(0)} d\Omega \tag{11}$$

If there is no scattered radiance, then

$$\Delta\Omega' = \int_{\Delta\Omega} \frac{v(\Omega)}{v(0)} d\Omega \tag{12}$$

where $\Delta\Omega'$ is the SVA $\Delta\Omega$ (see Appendices A, B).
If the contribution of the scattered radiance is small, then $\Delta\Omega' \cong \Delta\Omega$. When the
optical thickness is large or the forward scattering is dominant, the contribution of the
scattered radiances increases.
We estimate the magnitude of each term of the integrand:

$$\frac{v(\Omega) + \Delta\Omega C\bar{I}_{sca}(\Omega)}{v(0) + \Delta\Omega C\bar{I}_{sca}(0)} = \frac{v(\Omega) + \Delta\Omega C\bar{I}_{sca}(\Omega)}{v(0)(1 + \Delta\Omega C\bar{I}_{sca}(0)/v(0))} \tag{13}$$

Usually, the solar disk scan measurement is performed only when the scattered light is
much less than the direct solar irradiance:
$\Delta\Omega C\bar{I}_{sca}(0)/v(0) << 1$.
The magnitude of this term has already been estimated from the influence of the
scattered radiance in the field of view in the measurement of the sun-photometer; the
estimation error of the optical thickness due to the scattered radiance in the field of
view (Zhao et al. 2012, Sinyuk et al. 2012).
Equation (13) can be approximated as follows:


$$\frac{v(\Omega) + \Delta\Omega C\bar{I}_{sca}(\Omega)}{v(0) + \Delta\Omega C\bar{I}_{sca}(0)} \cong \frac{v(\Omega) + \Delta\Omega C\bar{I}_{sca}(\Omega)}{v(0)}(1 - \frac{\Delta\Omega C\bar{I}_{sca}(0)}{v(0)})$$

$$= \frac{v(\Omega) + \Delta\Omega C\bar{I}_{sca}(\Omega)}{v(0)}(1 - \varepsilon_3) \tag{14}$$

$$= \frac{v(\Omega)}{v(0)} + \frac{\Delta\Omega C\bar{I}_{sca}(\Omega)}{v(0)} - \frac{v(\Omega)}{v(0)}\varepsilon_3 - \frac{\Delta\Omega C\bar{I}_{sca}(\Omega)}{v(0)}\varepsilon_3$$

where
$$\varepsilon_3 = \frac{\Delta\Omega C\bar{I}_{sca}(0)}{v(0)}. \tag{15}$$

Therefore, eq. (11) is as follows.

$$\Delta\Omega' = \int_{\Delta\Omega} \frac{v(\Omega) + \Delta\Omega C\bar{I}_{sca}(\Omega)}{v(0) + \Delta\Omega C\bar{I}_{sca}(0)} d\Omega$$

$$\cong \Delta\Omega + \Delta\Omega \int_{\Delta\Omega} \frac{C\bar{I}_{sca}(\Omega)}{v(0)} d\Omega - \Delta\Omega\varepsilon_3 - \Delta\Omega \int_{\Delta\Omega} \frac{C\bar{I}_{sca}(\Omega)}{v(0)} d\Omega\varepsilon_3 \tag{16}$$

$$= \Delta\Omega\left\{1 + \int_{\Delta\Omega} \frac{C\bar{I}_{sca}(\Omega)}{v(0)} d\Omega - \varepsilon_3 - \varepsilon_3 \int_{\Delta\Omega} \frac{C\bar{I}_{sca}(\Omega)}{v(0)} d\Omega\right\}$$

Since $v(0) = CF_0$, the above eq. (16) becomes
$$\Delta\Omega' \cong \Delta\Omega\left\{1 + \int_{\Delta\Omega} \frac{\bar{I}_{sca}(\Omega)}{F_0} d\Omega - \varepsilon_3 - \varepsilon_3 \int_{\Delta\Omega} \frac{\bar{I}_{sca}(\Omega)}{F_0} d\Omega\right\} \tag{17}$$

$$= \Delta\Omega\left\{1 + \varepsilon_2 - \varepsilon_3 - \varepsilon_2\varepsilon_3\right\}$$

where
$$\varepsilon_2 = \int_{\Delta\Omega} \frac{\bar{I}_{sca}(\Omega)}{F_0} d\Omega \tag{18}$$

The fourth term is smaller than the second and third terms and it can be ignored. Then,
comparing the second and third terms in the parenthesis,
$$\varepsilon_2 = \int_{\Delta\Omega} \frac{\bar{I}_{sca}(\Omega)}{F_0} d\Omega = \int_{\Delta\Omega}\left\{\frac{1}{F_0} \cdot \frac{1}{\Delta\Omega} \int_{\Delta\Omega} I_{sca}(\Omega + \Omega')f(\Omega')d\Omega'\right\}d\Omega \tag{19}$$

$$\varepsilon_3 = \frac{\Delta\Omega}{F_0} \cdot \frac{1}{\Delta\Omega} \int_{\Delta\Omega} I_{sca}(0 + \Omega')f(\Omega')d\Omega'$$

$$\tag{20}$$

$$= \frac{\Delta\Omega\bar{I}_{sca}(\Omega = 0)}{F_0}$$

where $\varepsilon_2$ is the integral of the mean scattered light $\bar{I}_{sca}(\Omega)$ in the region of



$f(\Omega) > 0$, and $\varepsilon_3$ is the integral of scatted light in the FOV when facing toward the
sun.
The $f(\Omega)$ of the POM-02 consists of the core part from 0.0 to 0.5 degrees, which
takes large values, and the wing part from 0.5 to 2.5 degrees which takes small values.
Therefore, the integral can be written as follows.
$$\varepsilon_2 = \int_{\Delta\Omega} \frac{\bar{I}_{sca}(\Omega)}{F_0} d\Omega = \int_{\Delta\Omega(core)} \frac{\bar{I}_{sca}(\Omega)}{F_0} d\Omega + \int_{\Delta\Omega(wing)} \frac{\bar{I}_{sca}(\Omega)}{F_0} d\Omega \tag{21}$$

Since $\bar{I}_{sca}(\Omega) \approx \bar{I}_{sca}(\Omega = 0)$ in the core part, $\int_{\Delta\Omega(wing)} f(\Omega) d\Omega << 1$, and $\int_{\Delta\Omega(core)} d\Omega \cong \Delta\Omega$,
the first term of the integral $\varepsilon_2$ is as follows.
$$\int_{\Delta\Omega(core)} \frac{\bar{I}_{sca}(\Omega)}{F_0} d\Omega \cong \frac{\bar{I}_{sca}(\Omega = 0)}{F_0} \Delta\Omega . \tag{22}$$

This means that the integral of the core part in the integral $\varepsilon_2$ has the same
magnitude as $\varepsilon_3$ and the two terms offset each other, whereas the integral of the wing
part remains. The area of the integral of the wing part is larger than that of the core
part. Even if the integral of scattered light in the FOV is small compared to the solar
direct irradiance, the integral of the wing part becomes large and introduces errors in
the SVA estimation. That is, even if the measurement value of scattered light is
smaller than the direct sun measurement, $\bar{I}_{sca}(\Omega)\Delta\Omega / F_0 \approx 10^{-3}$, the integral of the
wing part becomes large:
$$\int_{\Delta\Omega(wing)} \frac{\bar{I}_{sca}(\Omega)}{F_0} d\Omega \approx \frac{\Delta\Omega(wing)}{\Delta\Omega} \times 10^{-3} \approx \frac{\Delta\Omega(wing)}{\Delta\Omega(core)} \times 10^{-3} = 2.4 \times 10^{-2} . \tag{23}$$

In this case, the magnitude of the error is about 2%.
Figures 2 and 3 show the values of $\varepsilon_2$ and $\varepsilon_3$ when the aerosol optical thickness
at 550 nm is changed. Here, the solar zenith angle is 30 degrees and the aerosol models
are the OPAC Continental average, Urban, and Desert types (Hess et al. 1998). The
simulation calculations of the scattered sky radiances were performed using the
subroutine in the SKYRAD package. The Ångström exponents of the Continental
average in the shorter (350 to 500 nm) and longer (500 to 800 nm) wavelength regions
are 1.11 and 1.42, respectively. Those of the Urban areas are 1.14 and 1.43, respectively,
and those of the Desert are 0.20 and 0.17, respectively.
When comparing $\varepsilon_2$ and $\varepsilon_3$, the signs are opposite and partially cancel out.



However, $\varepsilon_3$ is one order of magnitude smaller than $\varepsilon_2$, and thus $\varepsilon_2$ contributes to
the error in the calculation of the SVA. In the Continental average and Urban models,
if the aerosol optical thickness at 550 nm is less than 0.5, the second term $\varepsilon_2$ is less
than 0.5%, and if the aerosol optical thickness at 550 nm is less than 1, the second
term $\varepsilon_2$ is less than 1%. In the Desert model, which includes large particles, the
second term is less than 1% for shorter wavelengths, where desert particles have a
higher absorption than in the longer wavelength regions. However, even if the aerosol
optical thickness at 550 nm is less than 0.5, the second term is larger than 1% for some
wavelengths.
From these simulations, if the scattered light can be removed from the SVA
calculation, then an improvement in the accuracy of the calculations can be expected.
However, since the intensity of the scattered light depends on aerosol characteristics, it
is difficult to estimate the intensity of the scattered light from the measurements.
Furthermore, close to the sun, the value of scattered light cannot be measured due to
the direct sunlight. In POM-01 and POM-02, scattered light can only be measured
without being affected by direct sunlight at scattering angles of more than 3 degrees.
The SVA was calculated by subtracting the measurements for a scattering angle of 3
degrees and the accuracy of the estimation was examined. Although not shown in
detail, for the continental average and urban models, even if the aerosol optical
thickness is 2 at 550 nm, the error in the SVA estimation was less than 0.5%. This
indicates that if the measured value of scattered light can be subtracted, the
estimation accuracy of the SVA can be greatly improved.
From these results, when we determine the SVA by using the data from the solar
disk scan measurement, if the aerosol optical thickness is less than 0.5 and the aerosol
does not include large particles such as desert dust particles, the effect of the scattered
radiances on the SVA calculation is less than 0.5%, and $\Delta\Omega$ is well approximated by
$\Delta\Omega'$. Furthermore, if the measured value of the scattered light can be subtracted, the
estimation accuracy of SVA can be greatly improved.

**3.  SVA calculation with the SKYRAD package**
The software in the SKYRAD package is often used for SVA calculation from the data
of the solar disk scan measurement. However, the authors noticed that there are
problems in this program, and this section investigates these problems in detail.
In the measurement of the solar disk scan, a range of ±1 degree in the zenith angle
direction and ±1 degree in the azimuth direction relative to the sun in increments of
0.1 degrees is used, which produces a 21 × 21 grid with angular resolution of 0.1


degrees. Therefore, the data are taken from the sun for scattering angles of up to about
1.4 $(= (1\,\text{degree}) \times \sqrt{2}\,)$ degrees. As shown in Fig. 1, the influence of the direct solar
irradiance as a light source extends to about 2.5 degrees. To take this into
consideration, the integration is performed by extrapolation for angles larger than 1.4
degrees.

The following three problems exist in the SKYRAD package for calculating the SVA.

First, the data processing does not consider changes in the airmass (solar zenith

angle) during the solar disk scan measurement. However, in practice, if the solar disk
scan measurement is conducted when the airmass change (solar zenith angle) is small,
then the resulting error is also small. Also, this is not usually a problem unless the
measurement is conducted over an extended period of time.

Second, before starting the data processing, the minimum measured value is

subtracted from the measured values. As a result, the measurements of the scattering
angle between 1 and 1.4 degrees are greatly affected. By integrating the measured
value minus the minimum, the SVA is always underestimated, but the solution to this
problem is not straightforward.

Thirdly, the values between 1.4 and 2.5 degrees are not properly extrapolated.

Frequently, the extrapolated value does not decrease monotonically. In some cases, this
partially cancels out the underestimation of the integral.

In Fig. 4, an example of the integrand for the SVA calculation is shown. In the blue

curve with open squares, the minimum value is subtracted. This curve is then
integrated by the current SKYRAD program. Since the minimum value is subtracted,
the difference is noticeable at scattering angles greater than 1 degree. In this case, the
extrapolated value from 1.4 to 2.5 degrees is almost constant. In many cases, nearly
constant values were extrapolated as in this example. In some cases, the extrapolated
values increased. In the red curve with open circles, the minimum value is not
subtracted. The values between 1.4 and 2.5 degrees were extrapolated using the data
from 1.0 to 1.4 degrees. Considering Fig. 1, the decreasing trend is more realistic.

To investigate the differences in the calculation methods, several calculations were

performed.
The following steps in the calculations were varied,
(1)   Whether the minimum value was subtracted.
(2)   Whether the change in airmass was considered.
(3)   The method for the extrapolation in the range from 1.4 to 2.5 degrees.
(4)   Whether the horizontal cross-section of the FOV is assumed to be a circle or an



ellipse (the current SKYRAD package method uses an ellipse).
(5)   The method for determining the ellipse's parameters.
Data taken at MLO in October and November in 2015 were used in this study.
The solar disk scan measurement was made between 10:00 and 13:00 local time at
MLO. The optical thicknesses at wavelengths of 500 and 340 nm were at most 0.1 and
0.5, respectively. Therefore, the influence of the scattered light on the SVA calculation
is small.
The SAV was calculated for the six cases shown in Table 1, including Case 1, which is
the current method used by the SKYRAD package. In Cases 4, 5, and 6, the values in
the range 1.4 to 2.5 degrees were extrapolated as a linear function of the cosine of the
scattering angle. This linear function was determined by the least squares method
using the data with a scattering angle of more than 1 degree. The elliptic parameters
in Case 6 were determined by assuming that the shape of the FOV is a 2-dimensional
Gaussian distribution. The results of the comparison are summarized in Table 2.
The difference between Case 1 and Case 2 is whether or not the minimum value was
subtracted. Case 1, in which the minimum value was subtracted, results in an
underestimation of about 1 to 4%.
The difference between Case 2 and Case 3 is whether the change in airmass was
considered or not. The solar disk scan measurement was made between 10:00 and
13:00 local time at MLO. Therefore, the change in the air mass is less than 0.01, and
there was hardly any influence from the change in airmass.
The difference between Case 3 and Case 4 is the method of extrapolation used in the
range from 1.4 to 2.5 degrees. In the current SKYRAD package, the SVA was
overestimated by 0.6 to 2.1%.
Since there was hardly any influence from the change in airmass, in Case 1 and Case
4 the underestimation caused by the subtraction of the minimum value and the
overestimation caused by the poor extrapolation partially cancel each other out, and
the current SKYRAD package method underestimates the SVA by 0.5 to 1.9%.
The difference between Case 3 and Case 5 is whether the horizontal cross-section of
the FOV is assumed to be a circle or an ellipse. The difference between them was less
than 0.1%. This indicates that POM-02 was well tuned when it was shipped from the
manufacturer.
In Case 6, a different method for determining elliptic parameters from the current
SKYRAD package was used. Therefore, the difference between Case 4 and Case 6 is
the difference between the methods used to determine the elliptic parameters. There
was almost no difference between the current method and the new method. The





method used to determine the elliptic parameters thus has little effect on the SVA
estimation.

### 4. Annual trend and seasonal variation of SVA

Broadly speaking, the SVA is determined by the size of the pinhole and the focal
length of the lens. There is a possibility that these parameters may change with
degradation and the inside temperature. Therefore, the annual trend and seasonal
variation of the SVA are examined.
Figures 5 and 6 show the SVAs in the visible region (Si photodiode) and in the
near-infrared region (InGaAs photodiode) from 2008 and 2016, respectively. The
observation for the calibration at MLO was performed over about a month in October
and November every year. The lens in the visible region was replaced before the
observation in 2013.
In Fig. 5(a), time series of the SVA in channels 1 to 8 are shown for the SVA
calculated by the corrected method in this study. In Fig. 5(b), the SVA in channel 4 (500
nm) calculated by both the corrected and the current SKYRAD package methods are
shown for comparison. As stated in the above section, the SVA calculated by the
current method is lower than that calculated by the corrected one except for 2008.
Since the lens in the visible region was replaced before the calibration observation in
2013, it is difficult to investigate the annual trend of the SVA. Additionally, from this
figure, the accuracy of the SVA ((standard deviation)/mean) is estimated at about 1%
except in 2015.
From 2008 to 2012, the value of the SVA seems to be decreasing. The value of the
SVA in 2008 is larger than in other years. The values of the SVA are within ± 0.5%
except in 2008. From 2013 to 2016, the values of the SVA are within ±1%. The annual
variation of the SVA is less than or equal to the accuracy of the SVA. From these
results, the annual trend in the SVA cannot be seen in only 4 years of data, and even if
there is a trend, it is smaller than the measurement accuracy.
Figure 6(a) is the same as Fig. 5(a) except for channels 9 to 11 (1225, 1627, 2200 nm)
and Fig. 6(b) is the same as Fig. 5(b) except for channel 10 (1627 nm). In these
channels, the SVA calculated by the current method is also lower than that calculated
by the corrected one except in 2008.
The determination accuracy of the SVA is also estimated as about 1%. The lens in the
near-infrared region was not replaced in the period from 2008 to 2016. The trend in the
SVA cannot be seen in 8 years of data either. The values of the SVA in this period are
within ±1%, which is the determination accuracy of the SVA. From these results, the



annual trend of the SVA in the near-infrared channels cannot be seen in 8 years of data,
and even if there is a trend, it is smaller than the measurement accuracy.
Figure 7 shows the SVAs of POM-02 (Tsukuba) in the 500 and 1627 nm channels in
the period from January 2014 to December 2016. All data are plotted and the data are
scattered about ±2%, though the values in 2014 are a bit low. There is a large amount
of data in the winter, because there are many fine days in the winter in Tsukuba.
There are little data from spring to autumn and the data in the summer are scattered.
Since the estimated SVA is scattered, it is not possible to draw a clear conclusion, but
as can be seen from Fig. 7, the seasonal variation exceeding ±2% cannot be confirmed
in either channel. This also indicates that the temperature dependence of the SVA in
both detector regions cannot be seen. Since the data are taken over a short period of 3
years, no annual trend in the SVA can be detected.

**5. Summary and conclusion**
Atmospheric aerosols are an important constituent of the atmosphere. Measurement
networks covering an extensive area from ground and space have been developed.
SKYNET is a ground-based monitoring system using sky radiometers POM-01 and
POM-02 (Prede Co. Ltd., Japan). To improve the measurement accuracy, it is
important to know the characteristics of the instruments and calibrate them. There
are two constants that we must determine to make accurate measurements. One is the
calibration constant, and the other is the SVA of the radiometer.
In Part I, problems related to the estimation of the calibration constant were
investigated, and in Part II, problems related to the determination of the SVA of the
sky radiometer were described.
In this study, the data from two sky radiometers POM-02 of the JMA/MRI are
analyzed. One of the sky radiometers is used as a calibration reference, and the other
is used for the continuous measurement at the Tsukuba MRI observation site.
The FOV of POM-02 consists of a core part from 0 to 0.5 degrees and a wing part
from 0.5 to 2.5 degrees. The wing part is about 3 orders of magnitude smaller than the
core part, but the wing part contributes about 2% to the SVA.
A method for determining the SVA using the sun as a light source was proposed by
Nakajima et al. (1996). In this method, the radiance around the direction of the sun in
$0.1 \times 0.1$ degree increments is measured. These measurements include the scattered
light from aerosols and air molecules as well as the direct solar irradiance. These
scattered radiances cause errors in the SVA calculation.
The influence of the scattered light was evaluated by simulations. As a result, if the





aerosol optical thickness is less than 0.5 at a wavelength of 550 nm and the aerosol
does not include large particles such as desert dust particles, then the effect of the
scattered radiances on the SVA calculation is less than 0.5%. Furthermore, if the
measurements of the scattered light can be taken into account, the estimation accuracy
of SVA can be greatly improved.
The SKYRAD package for determining the SVA from the solar disk scan
measurements has several problems. The problems do not result in major errors in the
estimation of the SVA, but can cause a systematic underestimation.
First, the data processing does not consider the change in the airmass (solar zenith
angle) during the solar disk scan measurement. In practice, if the measurements are
taken over a period when the change in airmass is small, then there is almost no
problem. Second, before beginning the data processing, the minimum value is
subtracted from each measured value. This results in an underestimation of the SVA
by 1 to 4%. Thirdly, the values between 1.4 and 2.5 degrees are not properly
extrapolated. This overestimates the SVA value by 0.6 to 2.1%. Since the second and
third errors partially cancel each other out, if the current software is used, the error
will finally be an underestimation by 0.5 to 1.9%.
The annual trend in the SVA was examined using the data taken at MLO. Since the
optical thickness at a wavelength of 500 nm is 0.1 at most at MLO, the influence of the
scattered light is small. The accuracy of the SVA was estimated as about 1%. In the
visible region, the annual trend in the SVA cannot be seen in only 4 years of data from
2009 to 2012 and 2013 to 2016, and it is smaller than the measurement accuracy. In
the near-infrared region, the annual trend of the SVA cannot be seen in 8 years data
from 2008 to 2016, and it is smaller than the measurement accuracy.
The seasonal variation of the SVA was examined using the data taken at Tsukuba
from January 2014 to December 2016. Since the time series of the determined SVA was
scattered ±2%, it is not possible to draw a clear conclusion, but seasonal variation
exceeding ±2% could not be confirmed. Furthermore, as the temporal range of the data
was short, no annual trend could be detected.

**Acknowledgements**

This work was supported by the NIES GOSAT-2 project, Japan. This work was also
partially supported by JSPS KAKENHI Grant Number JP17K00531.


**Appendix A**





Let $f(\Omega)$ be the response function of the FOV, where $\Omega$ indicates the direction,
and when $\Omega = 0$, $f(\Omega = 0) = 1$.
The SVA is then as follows:
$$\Delta\Omega = \int_{\Delta} f(\Omega)d\Omega .$$      (A1)
Suppose parallel light enters from $\Omega = \Omega_0$.
$$V(\Omega = \Omega_0)$$
$$= C\int_{\Delta} f(\Omega)\delta(\Omega - \Omega_0)F_0 d\Omega$$      (A2)
$$= Cf(\Omega = \Omega_0)F_0$$

Therefore,
$$f(\Omega_0) = \frac{V(\Omega_0)}{CF_0} .$$      (A3)
Since $f(0) = 1$, then $V(0) = CF_0$.
Therefore,
$$\Delta\Omega = \int_{\Delta} f(\Omega)d\Omega$$
$$= \int_{\Delta} \frac{V(\Omega_0)}{CF_0} d\Omega_0$$      (A4)
$$= \int_{\Delta} \frac{V(\Omega_0)}{V(0)} d\Omega_0$$

When the parallel light is incident, the SVA of the radiometer can be obtained by
integrating the output in an arbitrary direction normalized by the output in the
direction of $\Omega = 0$.

**Appendix B**
Here, we consider the case that the light source has a finite size, for example, when
the sun is used as a light source.
Let the radiance distribution of the light source be $I(\Omega) = I_0 g(\Omega)$.
The integrated energy of the light source $F_0$ is as follows,
$$F_0 = \int_{\Delta} g(\Omega)I_0 d\Omega$$      (B1)
where $\Delta$ is the extent of the light source.
Considering the sun as a light source, let $\Delta$ be smaller than $\Delta\Omega$. Also, when the
sun is a light source, $F_0$ is the solar irradiance.
Let $C$ be the sensitivity of the detector, where $C$ is the proportional constant of





the sensor output and input energy.
The light source is in the direction of $\Omega = 0$ and we measure the radiance from it as

$$v(0) = C\int_{\Delta} f(0+\Omega')g(\Omega')I_0 d\Omega'$$


$$= CI_0\int_{\Delta} f(\Omega')g(\Omega')d\Omega' \qquad \text{(B2)}$$

where $v(0)$ is the sensor output.
If $f(\Omega)$ is constant within the range of $\Delta$ (POM-02 satisfies this condition), then
this equation can be rewritten as follows:

$$v(0) = CI_0\int_{\Delta} f(\Omega')g(\Omega')d\Omega'$$


$$= CI_0 f(0)\int_{\Delta} g(\Omega')d\Omega' \qquad \text{(B3)}$$

$$= Cf(0)F_0$$

$$= CF_0$$

Next, the light source is in the direction of $\Omega = \Omega_0$,
$$v(\Omega_0) = CI_0\int_{\Delta} f(\Omega_0+\Omega')g(\Omega')d\Omega' \qquad \text{(B4)}$$
where $v(\Omega_0)$ is the sensor output.
Then, both sides of the equation are integrated within the SVA $\Delta\Omega$,
$$\int_{\Delta\Omega} v(\Omega_0)d\Omega_0 = \int_{\Delta\Omega}\left(CI_0\int_{\Delta} f(\Omega_0+\Omega')g(\Omega')d\Omega'\right)d\Omega_0 \qquad \text{(B5)}$$
By changing the order of integration on the right, the following equation can be
obtained:

$$\int_{\Delta\Omega} v(\Omega_0)d\Omega_0 = CI_0\int_{\Delta}\left(g(\Omega')\int_{\Delta\Omega} f(\Omega_0+\Omega')d\Omega_0\right)d\Omega'$$


$$= CI_0\int_{\Delta} g(\Omega')d\Omega' \cdot \Delta\Omega \qquad \text{(B6)}$$

$$= CF_0\Delta\Omega$$

Therefore, from eqs. (B3) and (B6),

$$\Delta\Omega = \frac{1}{CF_0}\int_{\Delta\Omega} v(\Omega_0)d\Omega_0$$


$$= \int_{\Delta\Omega} \frac{v(\Omega_0)}{v(0)} d\Omega_0 \qquad \text{(B7)}$$

Thus, even in the case that the light source has a finite size, the SVA of the





radiometer can be obtained in the same manner as in the case of the parallel light
source.

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



Table titles
Table 1 Settings of the SVA calculation

Table 2 Influence of the different calculation settings.
(a) Calculated SVA. The data taken at MLO in October and November 2015 are used.
(b) Comparison of calculated SVA.

Figure captions
Fig. 1 Example of the measurement of the sun and the sky around the sun.
The measurement was performed keeping the same zenith angle as the solar zenith
angle. A negative (positive) scattering angle means the left (right) side of the
instrument is facing the sun. The red line with an open circle is the output of POM-02,
and the blue line is the output of the image sensor when shading the solar disk. Both
outputs are normalized by the value at a scattering angle of −3 degrees.

Fig. 2 Estimation of the error $\varepsilon_2$ in the calculation of the SVA. Aerosol models are the
OPAC Continental average, Urban, and Desert. The aerosol optical thickness is that at
a wavelength of 550 nm and the solar zenith angle is 30 degrees.

Fig. 3 Same as Fig. 2 but for error $\varepsilon_3$.

Fig. 4 Example of the integrand of the SVA calculation. The blue line with open
squares is for the case that the minimum value is subtracted, and the red line is for the
case that the values between 1.4 and 2.5 degrees are extrapolated using the data from
1.0 to 1.4 degrees.

Fig. 5 SVAs in the visible region (Si photodiode) from 2008 to 2016. The data were
taken at MLO over a month in October and November every year. (a) SVA calculated by
the corrected method in this study, (b) SVA at a wavelength of 500 nm calculated by
both the corrected and the current SKYRAD package methods.

Fig.6 Same as Fig. 5 but for the near-infrared region (InGaAs photodiode). The
wavelength in (b) is 1627 nm.

Fig.7 Time series of the SVA at POM-02 (Tsukuba) from January 2014 to December
2016: (a) 500 nm, (b) 1627 nm.




Table 1 Settings of the SVA calculation.

|  | subtract minimum value | consideration of airmass change | extrapolation method | FOV shape |
|---|---|---|---|---|
| case 1 | yes | no | current | elliptic |
| case 2 | no | no | current | elliptic |
| case 3 | no | yes | current | elliptic |
| case 4 | no | yes | new | elliptic |
| case 5 | no | yes | current | circular |
| case 6 | no | yes | new | elliptic |

Case 1 is the method implemented in the current SKYRAD package.
The elliptic shape parameters in Case 6 are calculated by a different method from the
SKYRAD package.





Table 2 Influence of the different calculation settings.

(a) Calculated SVA. The data taken at MLO in October and November 2015 are used.

| WLN (nm) | | 340 | 380 | 400 | 500 | 675 | 870 | 940 | 1020 | 1225 | 1627 | 2200 |
|---|---|---|---|---|---|---|---|---|---|---|---|---|
| Case_1 (C1) | SVA($\times 10^{-4}$) | 2.4495 | 2.4643 | 2.4472 | 2.4366 | 2.4530 | 2.4404 | 2.4554 | 2.4567 | 2.0086 | 2.0152 | 2.0692 |
| | SD($\times 10^{-4}$) | 0.0379 | 0.0407 | 0.0403 | 0.0388 | 0.0374 | 0.0277 | 0.0296 | 0.0241 | 0.0287 | 0.0241 | 0.0214 |
| | SD/SVA | 0.0155 | 0.0165 | 0.0165 | 0.0159 | 0.0153 | 0.0113 | 0.0121 | 0.0098 | 0.0143 | 0.0120 | 0.0103 |
| Case_2 (C2) | SVA($\times 10^{-4}$) | 2.5014 | 2.5186 | 2.5036 | 2.4764 | 2.4782 | 2.4995 | 2.5322 | 2.5564 | 2.0586 | 2.0737 | 2.1328 |
| | SD($\times 10^{-4}$) | 0.1151 | 0.1116 | 0.1144 | 0.0838 | 0.0579 | 0.0346 | 0.0314 | 0.0257 | 0.0294 | 0.0260 | 0.0233 |
| | SD/SVA | 0.0460 | 0.0443 | 0.0457 | 0.0338 | 0.0234 | 0.0138 | 0.0124 | 0.0101 | 0.0143 | 0.0125 | 0.0109 |
| Case_3 (C3) | SVA($\times 10^{-4}$) | 2.5015 | 2.5184 | 2.5035 | 2.4765 | 2.4783 | 2.4993 | 2.5320 | 2.5565 | 2.0586 | 2.0737 | 2.1327 |
| | SD($\times 10^{-4}$) | 0.1151 | 0.1115 | 0.1144 | 0.0838 | 0.0580 | 0.0344 | 0.0315 | 0.0258 | 0.0295 | 0.0260 | 0.0233 |
| | SD/SVA | 0.0460 | 0.0443 | 0.0457 | 0.0338 | 0.0234 | 0.0138 | 0.0124 | 0.0101 | 0.0143 | 0.0125 | 0.0109 |
| Case_4 (C4) | SVA($\times 10^{-4}$) | 2.4693 | 2.4899 | 2.4698 | 2.4534 | 2.4641 | 2.4691 | 2.4923 | 2.5023 | 2.0346 | 2.0440 | 2.1005 |
| | SD($\times 10^{-4}$) | 0.0668 | 0.0804 | 0.0698 | 0.0580 | 0.0459 | 0.0304 | 0.0302 | 0.0259 | 0.0301 | 0.0259 | 0.0227 |
| | SD/SVA | 0.0271 | 0.0323 | 0.0283 | 0.0236 | 0.0186 | 0.0123 | 0.0121 | 0.0104 | 0.0148 | 0.0127 | 0.0108 |
| Case_5 (C5) | SVA($\times 10^{-4}$) | 2.5027 | 2.5199 | 2.5032 | 2.4777 | 2.4783 | 2.5010 | 2.5329 | 2.5565 | 2.0596 | 2.0750 | 2.1336 |
| | SD($\times 10^{-4}$) | 0.1155 | 0.1123 | 0.1141 | 0.0831 | 0.0583 | 0.0346 | 0.0312 | 0.0262 | 0.0298 | 0.0261 | 0.0236 |
| | SD/SVA | 0.0461 | 0.0446 | 0.0456 | 0.0335 | 0.0235 | 0.0138 | 0.0123 | 0.0102 | 0.0145 | 0.0126 | 0.0111 |
| Case_6 (C6) | SVA($\times 10^{-4}$) | 2.4694 | 2.5042 | 2.4698 | 2.4535 | 2.4637 | 2.4698 | 2.4921 | 2.5028 | 2.0349 | 2.0449 | 2.1014 |
| | SD($\times 10^{-4}$) | 0.0669 | 0.1249 | 0.0701 | 0.0576 | 0.0463 | 0.0297 | 0.0305 | 0.0264 | 0.0312 | 0.0258 | 0.0225 |
| | SD/SVA | 0.0271 | 0.0499 | 0.0284 | 0.0235 | 0.0188 | 0.0120 | 0.0122 | 0.0106 | 0.0153 | 0.0126 | 0.0107 |



(b) Comparison of calculated SVA.

| WLN (nm) | 340 | 380 | 400 | 500 | 675 | 870 | 940 | 1020 | 1225 | 1627 | 2200 | |
|---|---|---|---|---|---|---|---|---|---|---|---|---|
| C2/C1-1 | 0.0212 | 0.0220 | 0.0230 | 0.0163 | 0.0103 | 0.0242 | 0.0313 | 0.0406 | 0.0249 | 0.0290 | 0.0307 | min. value subtraction |
| C3/C2-1 | 0.0000 | −0.0001 | 0.0000 | 0.0000 | 0.0000 | −0.0001 | −0.0001 | 0.0000 | 0.0000 | 0.0000 | 0.0000 | airmass change |
| C4/C3-1 | −0.0129 | −0.0113 | −0.0135 | −0.0093 | −0.0057 | −0.0121 | −0.0157 | −0.0212 | −0.0117 | −0.0143 | −0.0151 | different extrapolation |
| C4/C1-1 | 0.0081 | 0.0104 | 0.0092 | 0.0069 | 0.0045 | 0.0118 | 0.0150 | 0.0186 | 0.0129 | 0.0143 | 0.0151 | min. value subtraction, different extrapolation |
| C5/C3-1 | 0.0005 | 0.0006 | −0.0001 | 0.0005 | 0.0000 | 0.0007 | 0.0004 | 0.0000 | 0.0005 | 0.0006 | 0.0004 | circular or elliptic shape |
| C6/C4-1 | 0.0000 | 0.0057 | 0.0000 | 0.0000 | −0.0002 | 0.0003 | −0.0001 | 0.0002 | 0.0001 | 0.0004 | 0.0004 | different elliptic parameters |





Fig.1

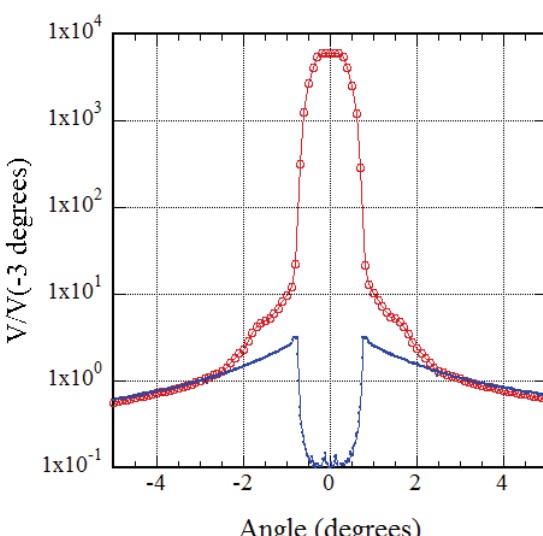

Fig. 1 An example of measurement of the sun and the sky around the sun. The measurement was performed keeping the same zenith angle as the solar zenith angle. A negative (positive) scattering angle means the left (right) side facing the sun. The red line with open circle is output of POM-02, and the blue line is output of image sensor output by shading the solar disk. Both output are normalized by the value at scattering angle -3 degrees





Fig. 2

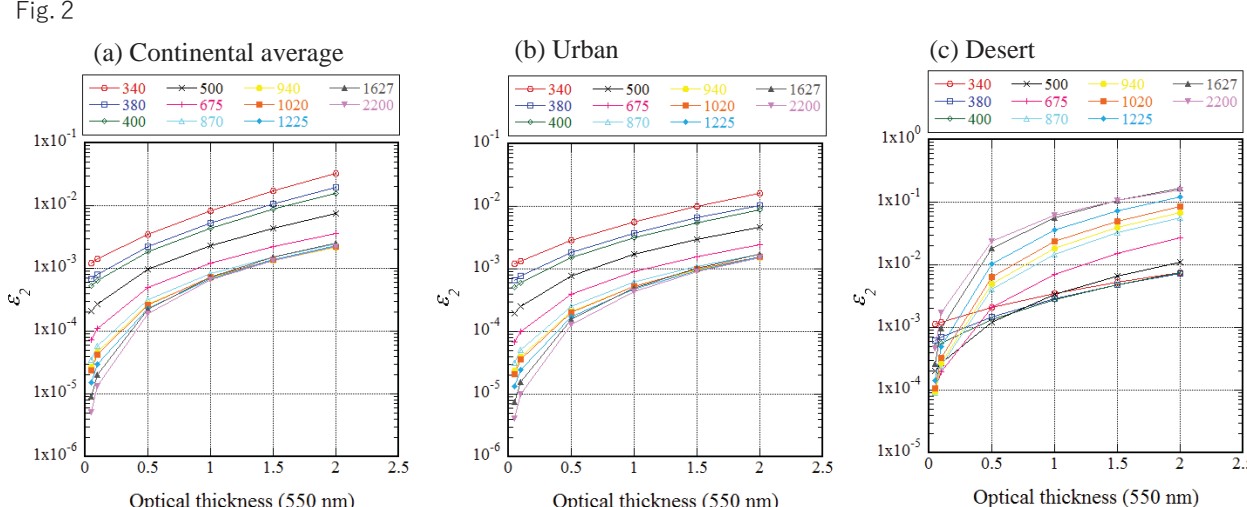

Fig. 2 Estimation of error $\varepsilon_2$ in calculation of SVA. Aerosol models are OPAC continental average, urban and desert. The aerosol optical thickness is that at the wavelength of 550nm and the solar zenith angle is 30 deg.




Fig. 3

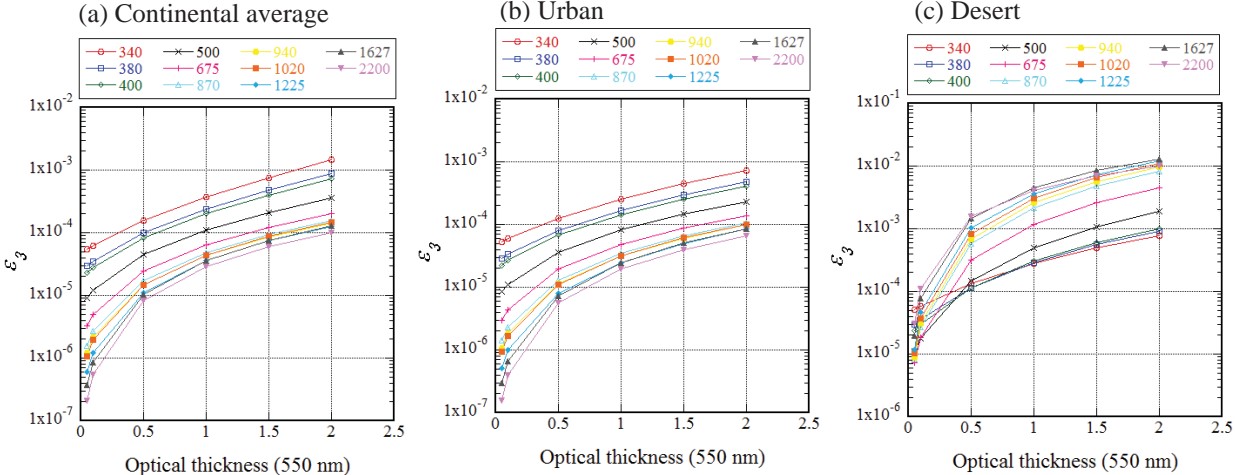

Fig. 3 Same as Fig. 2 but for error $\varepsilon_3$.



Fig.4

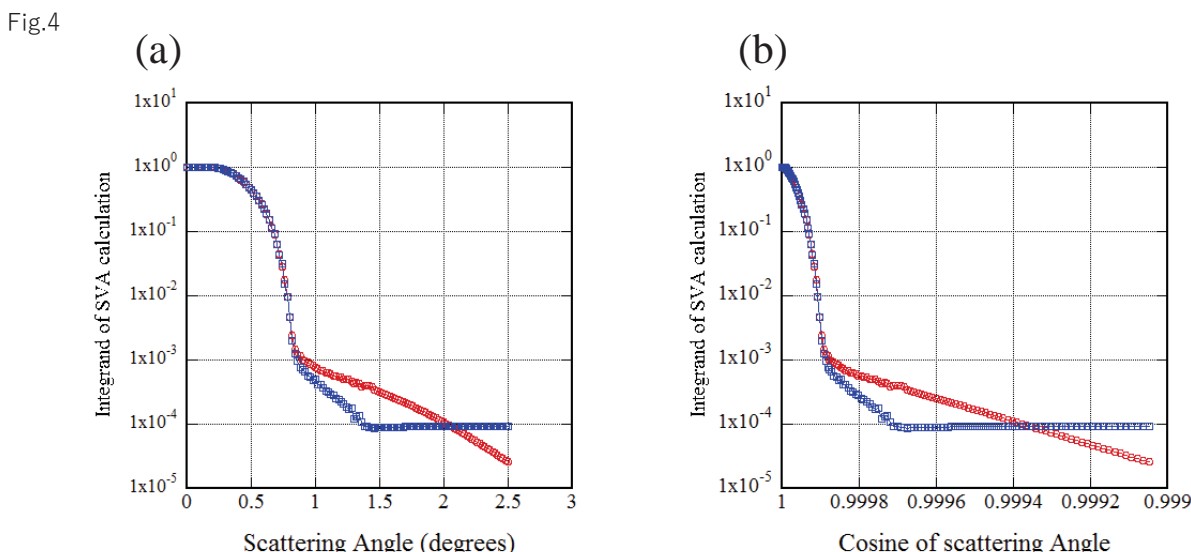

Fig.4 An example of integrand of SVA calculation. The blue line with open squares is the case that the minimum value is subtracted, and the red line is the case that the values between 1.4 and 2.5 degrees are extrapolated using the data from 1.0 to 1.4 degrees





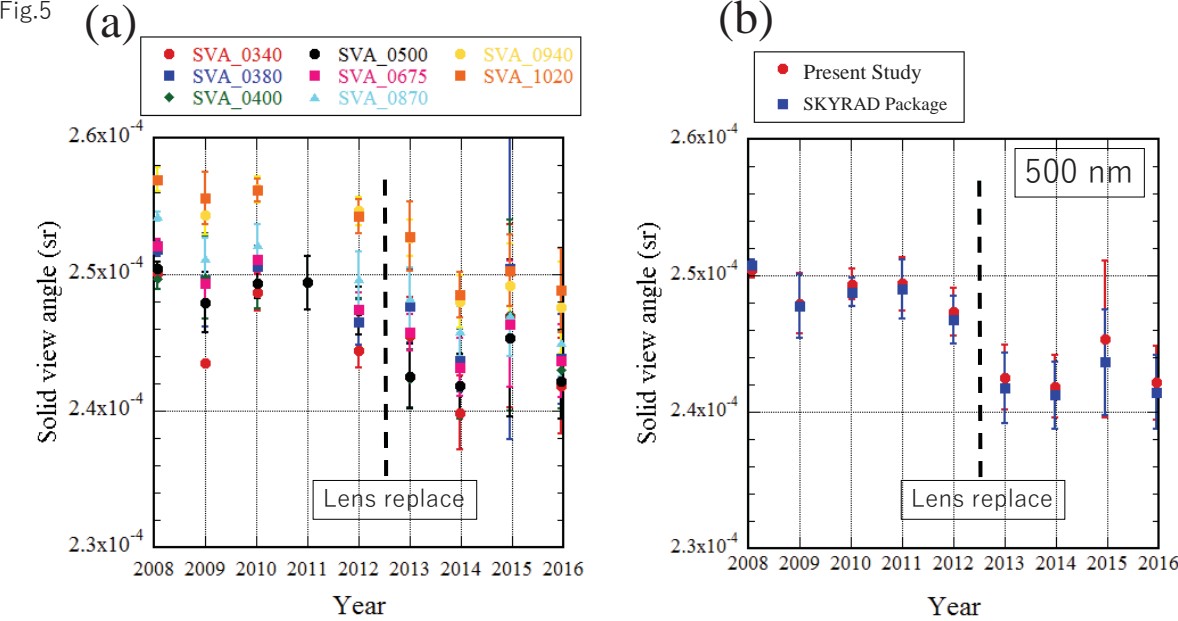

Fig.5 SVAs in the visible region (Si photodiode) from 2008 to 2016. The data were taken at MLO during about a month in October and November every year. (a) SVA calculated by the corrected method in this study, (b) SVA at the wavelength of 500 nm calculated by both the corrected and the current SKYRAD package methods.



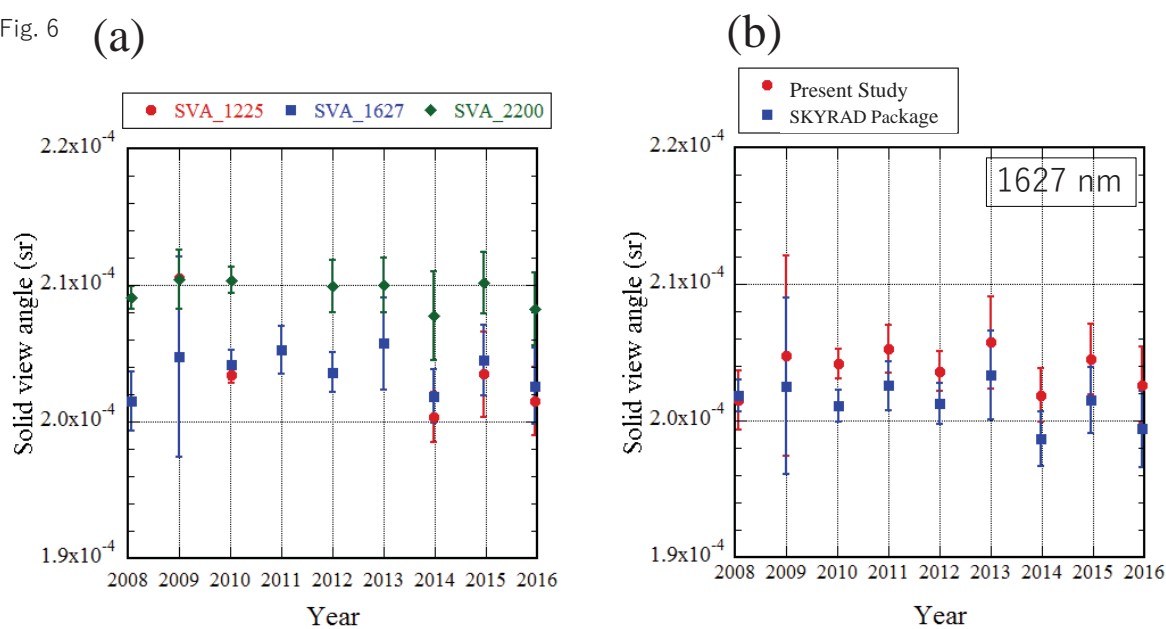

Fig.6 Same as Fig. 5 but for the near infrared region (InGaAs photodiode). The wavelength in (b) is 1627 nm.





Fig. 7

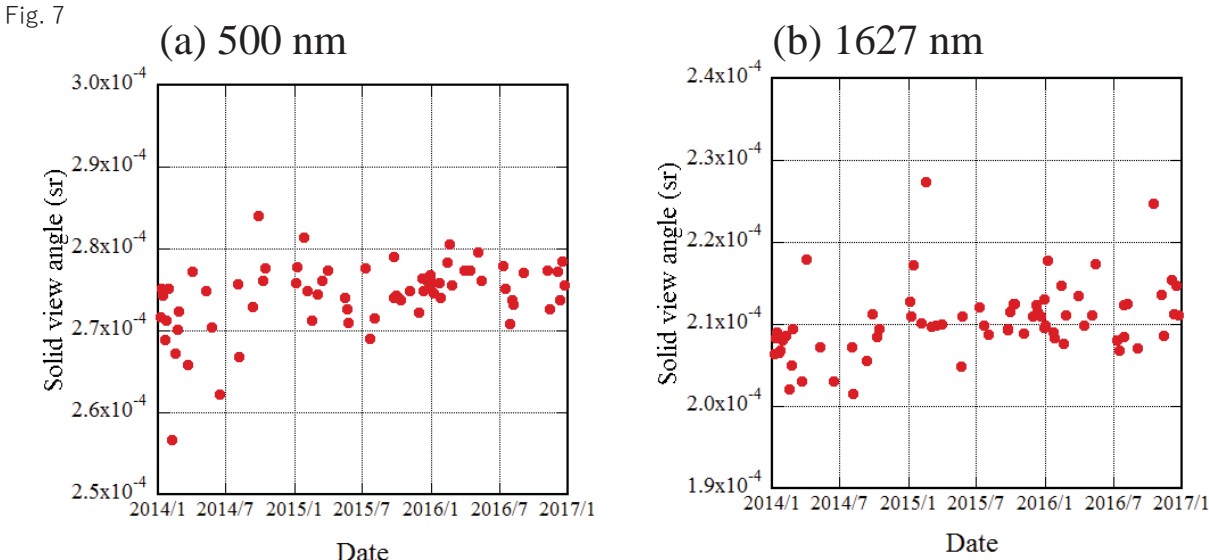

Fig.7 Time series of SVA of POM-02(Tsukuba) in the period from January 2014 to December 2016, (a) 500nm, (b) 1627nm.