# Peer review of "The instrument constant of sky radiometers (POM-02),"

_Atmospheric Measurement Techniques, 2017_

## Short Comment (SC1) · 19 Jan 2018

The article may prove to be very useful once revised as it contains many positive elements. At present the majority of methods and results are more summaries of summaries and approximations of approximations without references to the considerable work behind them.

For example, the primary argument that the f() function extends out to 2.5 degrees is based on a single paragraph outlining a summary of measurements of a an imaging sensor and its shading with no indication of how and what are the uncertainties of the imaging sensor data. 'The' f() described in the manuscript is one of many f(), and in this case the use of a finite sized object of ∼0.5 deg in diameter (the Sun); and neglects the

likelihood of a unfocussed image as suggested by fig 1. Also missing are the algorithm explanation of (a) correction for airmass, (b) the circular approximation (which could be simply matrix addition), and (c) a very light description of the new interpolation; none have references to referred articles. Similarly, terms are introduced (e.g. Fo) without explanation - usually after they have been used in equations. The Table 2(a) summary of the MLO data is interesting though there is no indication of the number of scans collected for each wavelength in the period Oct-Nov 2015. Nor why the (unbiased?) estimate of the standard deviations is lowest for the 'SkyRad' Case 1 compared to the others in some cases by a factor of 2 with no explanation for the increase for the wavelengths < 500 nm; other data from other workers suggests similar std dev across most wavelengths for low AOD (or AOT) locations. Therefore while the bias may (and one repeats may) have improved the estimate of the 'true' SVA the uncertainty of the mean increased.

Given the incorrect use of the term 'aerosol optical thickness' to represent aerosol optical depth in historical papers it would be useful if the authors could define their use of the term (e.g. is it AOT = AOD * M) in the paper particularly near line 288.

The positives of the article are many including the issues with SkyPak smoothing and extrapolation in Fig 4. Though the argument as to why the SkyPak minima is always around $10^{-4}$ (rather than just an algorithmic flaw in SkyPak) after scattering angle 1.4 is missing or why the divergence from the SkyPak f() occurs at about 0.8 deg when the extrapolation of the 'improved' method is implied to start at 1.4?
* * *

---

## Referee Comment (RC1) · Anonymous Referee #1 · 2 Apr 2018

The authors have carried out an analysis to quantify errors in the determination of the solid view angle of SKYNET (POM-02) radiometers. Because of the difficulty of this type of instrumental characterization, the analysis is almost entirely based on theoretical considerations. In general, the work represents a positive contribution that should be published after a few technical corrections.

Specific suggestions are:

- Improve documentation of the reported analysis methods by providing references to the original works.

- In the summary session add an statement on the accuracy of the current knowledge of SVA, and list the remaining sources of uncertainty, and ways of addressing in future

work.

---

## Editor Comment (EC1) · O. Torres (Editor) · 9 Apr 2018

The comment by Bruce Forgan provides valuable feedback for the improvement of the manuscript. I, therefore, recommend that the authors treat Bruce Forgan's short comment as an official RC review and reply to it accordingly.

Omar Torres

---

## Author Comment (AC1) · 1 May 2018

Reply to comments

We would like to thank you for reading our manuscript and commenting on it.
The comments are copied and shown below in italic.

*Comment.*
*"The article may prove to be very useful once revised as it contains many positive elements. At present the majority of methods and results are more summaries of summaries and approximations of approximations without references to the considerable work behind them."*
==>
Reply.
The method to calculate the SVA of skyradiometer POM-02 used in SKYNET is described in Nakajima et al. (1996), but we do not know other references. In the paper by Nakajima et al (1996), the description of the method is brief, and the details are unknown. Therefore, we summarized the theoretical basis in the Appendix by ourselves. Please let me know if you know the paper on how to determine the SVA of the skyradiometer with the sun as a light source.

*Comment.*
*"For example, the primary argument that the f0 function extends out to 2.5 degrees is based on a single paragraph outlining a summary of measurements of a an imaging sensor and its shading with no indication of how and what are the uncertainties of the imaging sensor data. 'The' f0 described in the manuscript is one of many f0, and in this case the use of a finite sized object of ~0.5 deg in diameter (the Sun); and neglects the likelihood of a unfocussed image as suggested by fig 1."*
==>
Reply.
Fig. 1 was shown to show the influence range of the direct solar irradiance in the measurement around the sun. We do not intend to discuss the measurement uncertainty of imaging sensor here.

We replace Fig. 1 with a new one. In the new Fig.1, the measurement examples at the wavelength 380 nm, 500 nm, and 675 nm of POM-02 are shown. The measurements were performed vertically at intervals of 0.1 degree scattering angles. Here, "vertically" means

that the measurements were performed while keeping the azimuth angle the same as the solar azimuth angle. The values are normalized by the measured value at the scattering angle zero (the direct solar irradiance).

We changed the paragraph from line 99 to 109 to the following one.

"An example of measurements of the radiance of the sun and around the sun is shown in Fig. 1. The measurements at POM-02 were performed vertically at intervals of 0.1 degree scattering angles, where the wavelengths are 380, 500, and 675 nm. Here, "vertically" means that the measurements were performed while keeping the azimuth angle the same as the solar azimuth angle. In Fig. 1, the values are normalized by the measured value at the scattering angle zero (the direct solar irradiance), where a positive (negative) value means the high (low) solar elevation side. At any wavelength, the output of POM-02 changes greatly around the scattering angle of −2.5 and 2.5 degrees. This means that the output of POM-02 is affected by the direct solar irradiance for up to about ±2.5 degrees from the sun direction. Fig. 1 also shows that scattering light is relatively larger as the wavelength is shorter."

We also added the following sentences after 105 lines.

"For ideal instruments, the output outside about 0.75 degrees should be the output due to scattering light by air molecules and atmospheric aerosols. However, Fig. 1 shows that the sensor output of POM-02 is affected by the direct solar irradiance for angles up to about ±2.5 degrees from the sun's center."

New Fig.1

[Figure]

Fig. 1 An example of measurement of the sun and the sky around the sun. The measurement was performed keeping the same azimuth angle as the solar azimuth angle. A positive (negative) value means the high (low) solar elevation side, where the wavelengths are 380 nm (red), 500 nm (blue), and 675 nm (green). The values are normalized by the measured value at the scattering angle zero (the direct solar irradiance).

*Comment.*

*"Also missing are the algorithm explanation of (a) correction for airmass, (b) the circular approximation (which could be simply matrix addition), and (c) a very light description of the new interpolation; none have references to referred articles. "*

==>

Reply.

(a) correction for airmass,

We insert the following sentences in 295 lines.

"In Cases 3, 4, 5, and 6, assuming that the aerosol optical depth has not changed, the solar direct irradiance changes due to the change of the airmass during the measurement."

(b) the circular approximation (which could be simply matrix addition),

In the SKYRAD package, we only assumed axisymmetric FOV. It seems unnecessary to explain in particular. We added the following sentence in Table 1.

"circular" means that the FOV is axisymmetric."

(c) a very light description of the new interpolation; none have references to referred articles.

From Fig.1, we think that linear extrapolation is appropriate. we do not think we need more detailed explanation.

In addition, Manago et al (2016) showed the results of FOV measured using a lamp in the 4th International SKYNET Workshop (Rome, March 2-4, 2016). According to the result, the FOV monotonically decreases between 1 degree and 2.5 degrees and then sharply decreases.

Here, the following sentences were added.

"Furthermore, Manago et al (2016) showed that the FOV monotonically decreases to around 2.5 degrees and then sharply decreases as the scattering angle increases on a lamp-based measurement on the ground."

Furthermore, the following reference was added.

"Manago, N, K. Pradeep, H. Irie, T. Takamura, and H. Kuze, 2016: On the method of solid view angle calibration for SKYNET skyradiometers, 4th International SKYNET workshop, Rome, March 2-4, 2016."

*Comment.*

   *"Similarly, terms are introduced (e.g. Fo) without explanation - usually after they have been used in equations. "*

==>

Reply.

   We added an explanation.

*Comment.*

   *"The Table 2(a) summary of the MLO data is interesting though there is no indication of the number of scans collected for each wavelength in the period Oct-Nov 2015. Nor why the (unbiased?) estimate of the standard deviations is lowest for the 'SkyRad' Case 1 compared to the others in some cases by a factor of 2 with no explanation for the increase for the wavelengths < 500 nm; other data from other workers suggests similar std dev across most wavelengths for low AOD (or AOT) locations. Therefore while the bias may (and one repeats may) have improved the estimate of the 'true' SVA the uncertainty of the mean increased. "*

==>

Reply.

   We entered the number of data in Table 2.

   Subtracting the minimum value from the measured values is a bug in the SKYRAD package. Therefore, it seems to be meaningless to compare the magnitude of the standard deviation of Case 1 with the other cases.

However, we insert the following sentences after 300 lines, if you request.

"The standard deviation in the region of shorter wavelength in Case 1 is smaller than the other cases.   One of the causes of the variation of the calculated SVA is caused by the variation of wing part of FOV. In the region of shorter wavelength, generally, the optical depth is thicker than the longer wavelength region, and the scattered light increases in the shorter wavelength region. When the minimum value is subtracted from the measurement value, the value of the wing portion decreases much in the shorter wavelength region, and the contribution to the SVA integration decreases much in the short wavelength region. As a result, the variance of the calculated SVA becomes small. However, there is no ground for subtracting the minimum value."

*Comment.*

*"Given the incorrect use of the term 'aerosol optical thickness' to represent aerosol optical depth in historical papers it would be useful if the authors could define their use of the term (e.g. is it AOT = AOD * M) in the paper particularly near line 288. "*
==>
Reply.

We replaced "optical thickness" with "optical depth".

*Comment.*

*"The positives of the article are many including the issues with SkyPak smoothing and extrapolation in Fig 4. Though the argument as to why the SkyPak minima is always around 10^-4 (rather than just an algorithmic flaw in SkyPak) after scattering angle 1.4 is missing or why the divergence from the SkyPak f() occurs at about 0.8 deg when the extrapolation of the 'improved' method is implied to start at 1.4? "*
==>
Reply.

The hood of POM-02 is designed so that the full field of view (FOV) is 1 degree. The size of the sun disk is about 0.5 degrees. Therefore, the direct solar irradiance can enter the detector for angles up to about 0.75 degrees from the sun's center. For ideal instruments, the output outside about 0.75 degrees should be the output due to scattering light by air molecules and atmospheric aerosols. However, Fig. 1 shows that the sensor output of POM-02 is affected by the direct solar irradiance for angles up to about ±2.5 degrees from the sun's center.

This the reason why the divergence from f() occurs at about 0.8 deg.

The magnitude of the sensor output between 0.75 and 2.5 degrees depends on the internal structure of the skyradiometer and the optical constant of the material. It happened that the magnitude was 10^-3 to 10^-4. If the contribution of this region to the SVA is large, the instrument should be repaired. Furthermore, as the optical depth increases, the scattering light increases and the minimum value is affected.

In the measurement of the solar disk scan, a range of ±1 degree in the zenith angle direction and ±1 degree in the azimuth direction relative to the sun in increments of 0.1 degrees is used, which produces a 21 × 21 grid with angular resolution of 0.1 degrees; the data are taken from the sun for scattering angles of up to about 1.4 ( = (1 degree) × $\sqrt{2}$ ) degrees. That is, 21 × 21 = 441 measurements are performed. This

measurement takes about 160 seconds.

If the sensor output between 0.75 and 2.5 degrees is sufficiently small, measurements with an azimuth angle of ± 1 degree and a zenith angle of ± 1 degree are sufficient, and changes in the airmass can be neglected if the measurement is performed near local noon. However, since the contribution to the integral of the wing part is about 2%, it is necessary to extrapolate between 1.4 degrees and 2.5 degrees and integrate it. There is a bug in the current SKYRAD package program, and it is not properly extrapolated, so another method is shown in this paper.

If a range of ±1.8 degrees in the zenith angle direction and ±1.8 degrees in the azimuth direction relative to the sun in increments of 0.1 degrees is used, $37 \times 37 = 1369$ measurements are performed and it takes about 480 seconds (6 minutes). In this case, changes in airmass cannot be neglect. Please note that $1.8 \times \sqrt{2} = 2.5$ here.

---

## Author Comment (AC2) · 1 May 2018

Reply to comments

We would like to thank you for reading our manuscript and commenting on it.
The comments are copied and shown below in italic.

*Comment.*

*"The authors have carried out an analysis to quantify errors in the determination of the solid view angle of SKYNET (POM-02) radiometers. Because of the difficulty of this type of instrumental characterization, the analysis is almost entirely based on theoretical considerations. In general, the work represents a positive contribution that should be published after a few technical corrections.*

*Specific suggestions are:*

*- Improve documentation of the reported analysis methods by providing references to the original works."*

==>

Reply.

The method to calculate the SVA of skyradiometer POM-02 used in SKYNET is described in Nakajima et al. (1996), but we do not know other references. In the paper by Nakajima et al (1996), the description of the method is brief, and the details are unknown. Therefore, we summarized the theoretical basis in the Appendix by ourselves. In the revised version, a flow chart is added to understand SVA calculation procedure in the SKYRAD package.

[Figure]

Fig. C1

SVA calculation Flow Chart

(1)

Solar Disk Scan (21x21 grid)

$R'$

$a$
$b$
$R$
$R'$
$a$

- Determination of center $(x_c, y_c)$ ($\sim(0,0)$)
- Determination of elliptic parameters
    direction of major axis of the ellipse
    ratio of minor axis to major axis ($b/a < 1$)
- Conversion of distance from $R'$ to $R$
    $R'_{max} \sim 1\text{deg} \times \sqrt{2} = 1.41\text{deg}$
- In the SKYRAD package, the min. value is subtracted
    from the measured values.

(2)

Measured values

Smoothing

(3)

Smoothed values

(4)

Interpolation

Extrapolation

- Integration
- Correction of ellipse

*Comment.*

*"- In the summary session add an statement on the accuracy of the current knowledge of SVA, and list the remaining sources of uncertainty, and ways of addressing in future work."*

==>

Reply.

   We will add the following sentences.

"According to the method based on the current measurement data, the precision is 1% in the high-altitude mountains area such as MLO and 1.5 to 2% in the low altitude area such as Tsukuba. The causes of the error may be an increase in the scattered light in the case of optically thick, a variation in the solar direct irradiance due to a change in the aerosol during the solar disk scan measurement, and an error in the pointing direction of FOV. In the future we will eliminate scattered light and use measurements of aerosol optical depth by other instruments during the solar disk scan measurement. We also develop methods for measuring SVA on the ground or laboratory."